# Efficacy of a Validated Yoga Protocol on Dyslipidemia in Diabetes Patients: NMB-2017 India Trial

**DOI:** 10.3390/medicines6040100

**Published:** 2019-10-11

**Authors:** Raghuram Nagarathna, Rahul Tyagi, Gurkeerat Kaur, Vetri Vendan, Ishwara N. Acharya, Akshay Anand, Amit Singh, Hongasandra R. Nagendra

**Affiliations:** 1Swami Vivekananda Yoga Research Foundation, Bengaluru 560105, India; vetriyoga@gmail.com (V.V.); dramits90@gmail.com (A.S.); chancellor@svyasa.edu.in (H.R.N.); 2Neuroscience Research Lab, Department of Neurology, Postgraduate Institute of Medical Education and Research, Chandigarh 160012, India; rahul15tyagi@gmail.com (R.T.); Kaurgurkeerat6@gmail.com (G.K.); 3Central Council for Research in Yoga & Naturopathy (CCRYN), Delhi 110058, India; acharyaishwar@gmail.com

**Keywords:** diabetic yoga protocol, DYP, dyslipidemia, T2DM, diabetes mellitus

## Abstract

**Background:** Dyslipidemia is considered a risk factor in Type 2 diabetes mellitus (T2DM) resulting in cardio-vascular complications. Yoga practices have shown promising results in alleviating Type 2 Diabetes pathology. **Method:** In this stratified trial on a Yoga based lifestyle program in cases with Type 2 diabetes, in the rural and urban population from all zones of India, a total of 17,012 adults (>20 years) of both genders were screened for lipid profile and sugar levels. Those who satisfied the selection criteria were taught the Diabetes Yoga Protocol (DYP) for three months and the data were analyzed. **Results:** Among those with Diabetes, 29.1% had elevated total cholesterol (TC > 200 mg/dL) levels that were higher in urban (69%) than rural (31%) Diabetes patients. There was a positive correlation (*p* = 0.048) between HbA1c and total cholesterol levels. DYP intervention helped in reducing TC from 232.34 ± 31.48 mg/dL to 189.38 ± 40.23 mg/dL with significant pre post difference (*p* < 0.001). Conversion rate from high TC (>200 mg/dL) to normal TC (<200 mg/dL) was observed in 60.3% of cases with Type 2 Diabetes Mellitus (T2DM); from high LDL (>130 mg/dL) to normal LDL (<130 mg/dL) in 73.7%; from high triglyceride (>200 mg/dL) to normal triglyceride level (<200 mg/dL) in 63%; from low HDL (<45 mg/dL) to normal HDL (>45 mg/dL) in 43.7% of T2DM patients after three months of DYP. **Conclusions:** A Yoga lifestyle program designed specifically to manage Diabetes helps in reducing the co-morbidity of dyslipidemia in cases of patients with T2DM.

## 1. Introduction

### 1.1. Prevalence, Burden of Type 2 Diabetes Mellitus and Dyslipidemia

Type 2 diabetes mellitus (T2DM) is a chronic hyperglycemic metabolic disorder that occurs due to a complex interaction of several prevailing lifestyle factors including diet, obesity, and physical inactivity against a background of genetic and epigenetic factors [1]. These contribute to insulin resistance and relative insulin deficiency over time [2], resulting in complications such as cataracts, retinopathy, neuropathy, and nephropathy [3].

Recent worldwide findings show that the prevalence of Diabetes in adults in developing countries when compared to developed countries has increased from 4.3% to 9.0% in men and 5.0% to 7.9% in women in the last 35 years [4]. In India, it has been estimated that around 66.8 million people have diabetes [5]. As expected, not until very recently was the prevalence in the rural population known. A recent study of rural dwelling reported that 8.03% of the recruited population from western Uttar Pradesh was suffering from Diabetes and amongst them 9.91% were females and 6.79% were males [6]. Interestingly, a study assessed the mean expenditure in the management of T2DM and revealed the Indian rupee 853.47 to be the estimated average monthly expenses per individual. This expenditure increases with the duration of the disease and is concomitant with an increase in the complications associated with it [7]. In light of the increased expenditure on screening, diagnosis, monitoring, and management of T2DM, the combined health budget of developing countries is affected [5]. Diabetes and pre-diabetes together affects almost one fourth of the Indian population, which is clearly traceable to the rapidly changing lifestyle [8,9]. 

Hypertension and hyperlipidemia are the most common co-morbidities associated with T2DM. Both these co-morbidities hasten the occurrence of renal and cardiovascular complications at an alarming rate [10]. Myocardial infarction remains the most common cause of death in T2DM patients [11,12] as well as the prime cause of deaths worldwide [10,13]. Briefly, the factors responsible for an increase in cardio-vascular diseases (CVD) are diabetes, hypertension, abnormal cholesterol and triglyceride levels, high levels of low density lipoprotein (LDL), low levels of high density lipoprotein (HDL), and obesity [14]. Murray and Lopez predicted that between 1990–2020, CVD will cause devastating effects on human health and as expected, the diabetes population with dyslipidemia will be severely affected [15]. A cross sectional study conducted on 297 Indian subjects reported a direct correlation between the blood levels of glycated hemoglobin (HbA1c) and total cholesterol, LDL, and triglycerides besides an inverse relation with HDL levels [16,17]. Furthermore, it was shown that T2DM associated with decreased HDL and increased production of triglycerides, LDL, and very low lipoproteins (VLDL) levels is presumably due to impaired catabolism [18]. 

### 1.2. Lifestyle Interventions

Studies continue to reveal that even though T2DM is widely prevalent in India, there is a lack of access to healthcare, awareness, counseling, and treatments. Studies have recommended an improvement in lifestyle for managing all modifiable risk factors for the secondary prevention of complications in patients with T2DM [19]. 

### 1.3. Why This Yoga Study

In pursuit of better and cost-effective intervention of T2DM, Yoga has emerged as a natural and widely therapeutic option. Supporting evidence has shown that Yoga can positively impact T2DM patients by reducing fasting blood glucose (FBG) and HbA1c levels, and helps in improving the lipid profile and hypertension of T2DM patients [20]. It has been shown that Yoga has re-emerged as an integrated approach and found to be effective in the prevention and treatment of T2DM and dyslipidemia to the extent that it reduces overall healthcare costs by reducing future medications and hospitalizations [21,22]. However, a standardized protocol based on a Yoga lifestyle research is needed for effective community health translation [23,24]. Therefore, we examined the effect of the Diabetes Yoga lifestyle protocol on the lipid profile of a large sample population of patients with T2DM across India.

## 2. Materials and Methods

### 2.1. Study Design

This was a stratified translational research study in randomly selected cluster populations from all zones of rural and urban India. Study was approved by the Institutional Ethics Committee of the Indian Yoga Association (IYA) vide Res/IEC-IYA/001 dated 16 December 2016. The current study was registered in the Clinical Trial Registry of India (CTRI) vide CTRI/2018/03/012804. 

### 2.2. Screening and Recruitment of Participants

The study participants were recruited after obtaining signed informed consent as per the guidelines outlined by the Institutional Ethical Committee of Indian IYA. Detailed methodology adopted has been previously reported [25,26]. In brief, baseline assessment was carried out by a nationwide door to door screening in urban and rural districts under the flagship of the *Niyantrita Madhumeha Bhārat (NMB) Abhiyān (Diabetes Control Program),* funded by the Ministry of Health and Family Welfare, the ministry of AYUSH, Government of India, New Delhi, and conducted by IYA. The Diabetes Yoga Protocol (DYP) was validated by the Quality Council of India. Subjects were recruited based on the Indian Diabetic Risk Score (IDRS). T2DM individuals and those with a high risk on the Indian diabetes risk score, detected during screening at the randomly selected clusters of villages (rural) and census enumeration blocks (urban), were recruited and enrolled. They adopted a three months Yoga lifestyle protocol under the supervision of a trained Yoga teacher. Subjects adhered to five days a month for three months and attendance was recorded. Compliance was also ensured by WhatsApp reminders for the Yoga sessions.

### 2.3. Selection Criteria

All subjects who satisfied the selection criteria and signed the informed consent were included in the study. Medication to T2DM was not considered to be exclusion criteria. However, detailed medication records were not available. Known/self-reported/newly detected Diabetics or high risk subjects of all genders between age ranges of 20–70 were selected for analysis. For newly detected T2DM, the HbA1c level (6.5%) in high risk (>60 on IDRS) population was considered. Subjects with other co-morbidities including cancer, serious cardiac illness, chronic liver, pulmonary, neurological, renal diseases, lower back pain, and surgical interventions were excluded. Willingness to register in the trial was mandatory for inclusion.

### 2.4. Assessments

All assessments were carried out at the baseline and after three months of DYP intervention.

### 2.5. Anthropometric Assessments

Anthropometric assessments included height, weight, waist circumference, and hip circumference carried out at the time of screening of the subjects and at follow up.

### 2.6. Biochemical Assessments

Biochemical assessments included fasting blood glucose, glycated hemoglobin (HbA1c), total cholesterol, triglycerides, LDL, VLDL, and HDL. Assessments were carried out by accredidated diagnostics lab using standard diagnostic tools and procedures acceptable for public utility.

### 2.7. Intervention

DYP was designed by the Delphi method and focused group discussion by experts from Yoga traditions of the Indian Yoga Association and researchers on Diabetes [26]. The practices were taught by a certified Yoga instructor volunteers in nine day camps (2 h daily) in their respective villages or wards. Subsequently, they were asked to continue the practices daily (one hour) at home, through the use of DVDs. Weekly follow up classes were conducted at the same venues for three months. The detailed methodology is provided in Appendix A.

### 2.8. Statistical Analysis

The statistical analysis was carried out using SPSS 21.0 software (IBM Corp., Armonk, NY, USA). The normality was tested by using the Kolmogorov–Smirnov test. The comparisons were made by using paired samples t-test for normally distributed data. Significance level of various proportions were analyzed by the Chi square test. McNemar’s test was performed to assess the conversion. *p* value < 0.05 was considered to test the level of significance. 

## 3. Results

In this nationwide study, a total of 17,012 participants with high risk on the IDRS (>60) and known Diabetes were recruited. Out of the total, 5150 had HbA1c > 6.5%. Data was collected at two time points (i.e., before and after the Yoga intervention). Out of the 5150 T2DM patients with HbA1c >6.5%, 1745 individuals were found to have serum total cholesterol (TC) level > 200 mg/dL (borderline and hypercholesterolemia range). Post data on cholesterol values were available for analysis in 694 individuals after three months of a Yoga lifestyle regime. In the absence of medication details, the comparisons were carried between ≥200 and <200 cases with T2DM. Detailed study profile has been provided in Figure 1.

### 3.1. Effect of Diabetes Yoga Protocol on Diabetes Population with Dyslipidemia

The DYP reduced the TC levels in 4% of T2DM patients with TC > 200 mg/dL; TC reduced significantly (*p* ≤ 0.001; *t* = 22.93) from 232.34 ± 31.48 (pre yoga intervention) to 189.38 ± 40.23 (post yoga intervention). Other variables of lipid profile including triglycerides (Tg), LDL, and VLDL were also found to be significantly reduced after Yoga intervention. Interestingly, the HDL increased significantly in those with low (<45 mg/dL) baseline values and decreased significantly in those with high (>45 mg/dL) baseline values (Table 1). Total cholesterol reduced significantly in all age groups.

### 3.2. DYP Is Beneficial for TC in Diabetes of All HbA1c Categories in Both Genders of Urban and Rural Population

#### 3.2.1. Dyslipidemia Higher in Rural Population

Total cholesterol values were >200 mg dL in 29.1%; of these TC was significantly higher in the rural diabetes population (31%) than their urban counterparts (28%) (Appendix A). However, after DYP intervention, the reduction in hyperlipidemia was significantly better (*p* < 0.001.) in the rural diabetes population than in urban areas (5% vs. 3%, respectively).

#### 3.2.2. DYP Reduces Dyslipidemia Equally in Both Genders

The percentage of T2DM subjects with TC level >200 mg/dL was less when compared to those with cholesterol levels <200 mg/dL in both genders. Mean TC level was found to be similar in both the genders with T2DM (Appendix A).

#### 3.2.3. Hyperlipidemia Increases with Increasing HbA1c

The data showed that the levels of HbA1c and mean blood glucose levels were found to be positively correlated (*p* = 0.048) with the TC levels. This reveals that, with the increase in HbA1c levels, the percentage of T2DM subjects with cholesterol level > 200 mg/dl increased gradually (Figure 2). Moreover, after yoga intervention, the cholesterol levels were significantly reduced in all HbA1c categories (Appendix A).

#### 3.2.4. Effect of DYP on the Conversion of Hyperlipidemia in Diabetes Patients

A highly significant percentage of patients with Diabetes who had abnormal lipid levels were converted to normal levels after three months of yoga practice: 60.3% in total cholesterol, 73.7% in LDL, 63% in triglycerides, and 43% in those with low HDL (45 mg/dL) (Table 2). 

## 4. Discussion

### 4.1. Summary

We assessed the effect of a validated yoga lifestyle protocol on the lipid profile of 5150 patients with T2DM. There was a positive correlation between cholesterol with HbA1c values. A higher proportion of rural subjects, diabetic females with high A1c, and urban patients with high A1c were found to have higher cholesterol levels at the baseline, indicating vulnerability to serious complications.

Exposure to DYP resulted in a significant reduction in total cholesterol, LDL, VLDL, and triglyceride levels. However, HDL was the least affected in the rural region. DYP reduced the cholesterol levels, better in males than females; and DYP was equally beneficial in all age groups in both the urban and rural population in different ranges of HbA1c levels. 

### 4.2. Comparisons

Prior to the current study, Shantakumari et al. evaluated the impact of three months Yoga intervention on the dyslipidemic profile in 100 T2DM subjects and found similar findings [27]. However, the sample size (n-100) was smaller and represented only a south Indian population whereas the current pan-India study had a large sample size (5150) using a validated common yoga protocol. The outcomes of this DYP study showed an improvement in overall lipid profile (i.e., decrease in total cholesterol, triglycerides and LDL levels) among the T2DM subjects. Similarly, Mohammed et al. reported reduced total cholesterol, triglycerides, and LDL cholesterol in 158 Yoga practicing Type 2 Diabetes and dyslipidemia patients in comparison to the sulphonyl urea treatment group; the mean TC of 240.36 mg/dL (High) was reduced to 214.11 mg/dL (borderline) after four months of yoga intervention with a 10% reduction [28]. However, in our study, DYP intervention helped hyperlipidemia subjects to attain normal levels (189.38 mg/dl) of cholesterol from the baseline levels of 232.34 mg/dL with an 18% reduction. A recent systemic review of controlled Yoga trials on adult Diabetics recommended additional high quality studies due to methodological limitations in previous studies [29]. A recent meta-analysis reporting significant improvements in the lipid profile remained limited to non DYP protocols [30], highlighting the importance of validated protocols to further our deeper mechanistic understanding while retaining the reproducibility. Therefore, a specific Diabetes Yoga Protocol was employed in this nationwide study focused on T2DM patients of all age groups across India. We found a positive correlation (*p* < 0.05) between increasing levels of HbA1c and TC (Figure 2). Earlier studies have also shown similar results with high HbA1c as an important predictor of high serum lipid levels in T2DM subjects [17,31,32], warranting glycemic control as an important factor needed to control dyslipidemia and prevent major cardiovascular events [33].

### 4.3. Mechanism of action of Yoga

Yoga mediated reduction in the dyslipidemia has yet not been explored to the current scale in the Diabetic subjects. We describe the underlying mechanism based on existing studies. Mechanistically, insulin resistant cells inhibit lipase activity, the enzyme that catabolizes the lipids resulting in increased triglycerides, LDL, and cholesterol levels in the body [32]. In addition, the accumulation of lipids increases the risk of other co-morbidities like atherosclerosis, cardiovascular, and coronary artery diseases. Dyslipidemia also causes endothelial damage, which results in the loss of physiological vasomotor activity [34]. Furthermore, factors like dyslipidemia also contribute to increased blood pressure [35], which leads to the activation of the RAAS pathway where the aldosterone hormone is secreted due to over activation of the HPA axis in T2DM subjects [36]. Available evidence shows that Diabetes neuropathy affects the longest fiber of the parasympathetic system, leading to sympathetic imbalance, thus leading to hypertension [37]. However, the existing evidence also indicates that there is a persistent increase in the HPA (hypothalamus pituitary adrenal) axis activity in Diabetic patients with Diabetes neuropathy [38]. A number of studies depict that there is mitochondrial dysfunction and decreased activity of mitochondrial enzymes in T2DM subjects due to insulin resistance. Measurement of oxidative phosphorylation in vivo by P-NMR has also shown impaired ATP synthesis in insulin resistant subjects [39]. Furthermore, PPARδ (peroxisome proliferate activator receptor) is a lipid activated nuclear factor that has an important role in the regulation of glucose, lipid, and lipoprotein metabolism. Pre-clinical evidence has shown that PPAR can reduce or prevent obesity induced insulin resistance and T2DM [40]. Moreover, PPAR agonists are believed to be potent activators of lipid metabolism, thus explaining its beneficial actions on insulin sensitivity and adiposity [41]. Regardless of this pharmacological context, it has also been described that Yoga improves the lipid profile in T2DM subjects by increasing hepatic lipase and pancreatic lipase activity [27,42]. It has also been described that Yoga helps to maintain a balance between sympathetic and parasympathetic balance [43]. Specifically, the *Pranayama* practices (as part of most of Yoga protocols), also included in the DYP, are believed to decrease the blood sugar level by increasing the utilization and mechanism of glucose in liver adipose tissue and peripheral organs [44]. Blood supply to muscles is also improved with *Pranayama*, which enhances the insulin receptor expression in muscles and increases the glucose uptake by cells, thus reducing the blood sugar levels [45]. We argue that asanas included in the DYP may improve the accessibility of various enzymes to target and stimulate their substrates. This might contribute toward the reduction of LDL and TG.

### 4.4. Limitations

The study duration was one of the limitations as the analysis could not extend beyond three months. Yet, the drop off rate in the study could be ascribed to challenges in adopting a Yoga lifestyle, general laziness, inability to perform Yoga due to health limitations, and relative interest in other forms of physical exercises. In certain places, climatic and political conditions also led to drop outs.

### 4.5. Strengths

One of the strengths of this study was the inclusion of a large sample size and the use of a validated Diabetes Yoga Protocol. Longitudinal studies may examine the long term effects of DYP.

## 5. Conclusions

DYP significantly attenuated the hyperlipidemic state of T2DM patients. The potential of DYP to halt the conversion of hyperlipidemic into CVD among Diabetics can be probed by a longitudinal intervention study. There is not only a need to understand the mechanism governing the effects of DYP, but also in scaling it into a public intervention national program. Although the available evidence proves the significance of the beneficial impact of Yoga on the cholesterol levels, Tg, LDL, and VLDL, a standardized approach may further alleviate the fatal consequences of Diabetes. This may reduce vulnerability to heart diseases.

## Figures and Tables

**Figure 1 medicines-06-00100-f001:**
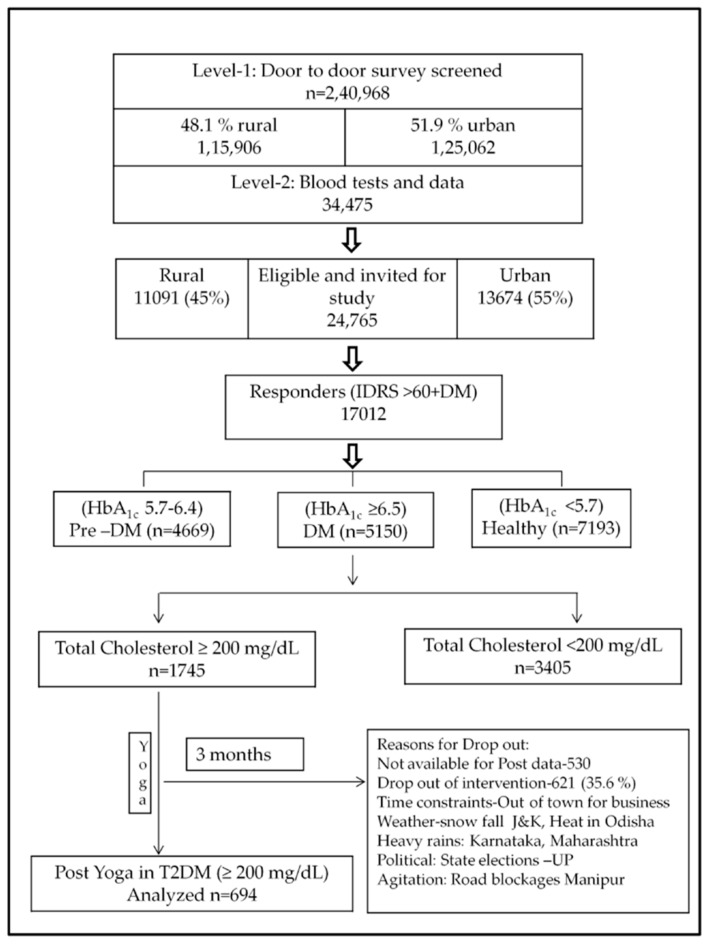
Study profile.

**Figure 2 medicines-06-00100-f002:**
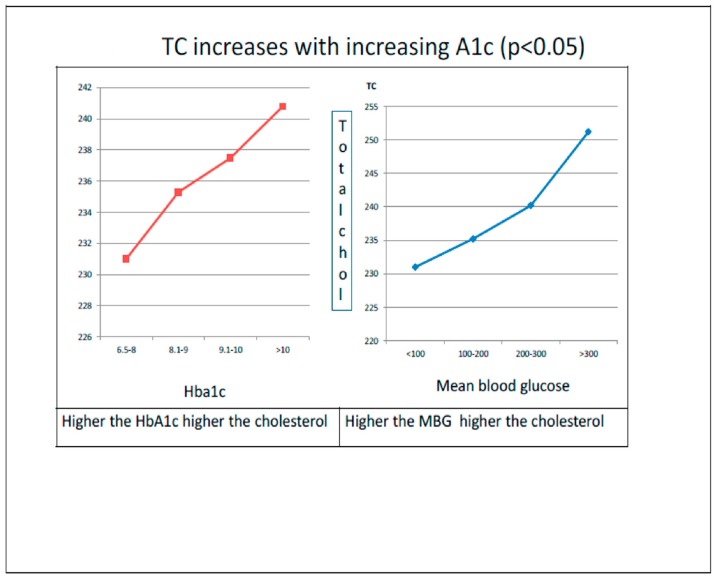
Relationship of HbA1c and hyperlipidemia.

**Table 1 medicines-06-00100-t001:** Pre post changes in lipids in those with high Total Cholesterol in type 2 diabetes patients in different age groups.

		MBG	TC	Tg	LDL	VLDL	HDL <45 mg/dL	HDL >45 mg/dL	Cho:HDL Ratio	LDL:HDL Ratio
		Mean (SD)	Mean (SD)	Mean (SD)	Mean (SD)	Mean (SD)	Mean (SD)	Mean (SD)	Mean (SD)	Mean (SD)
Overall	pre	152.3 (64.3)	239.4 (31.6)	201.0 (117.5)	142.0 (31.3)	35.7 (15.0)	34.9 (4.92)	55.1 (13.5)	4.66 (1.35)	2.91 (1.21)
	post	142.8 (52.7)	189.8 (40.5)	173.6 (98.8)	108.4 (34.9)	30.9 (14.9)	44.2 (12.0)	49.1 (11.8)	4.03 (1.24)	2.26 (0.94)
20–30 years	pre	143.87 (51.05)	230.35 (27.04)	204.90 (109.25)	140.03 (28.33)	35.95 (15.37)	42.7 (10.7)	61.0 (17.3)	4.55 (1.44)	2.80 (1.07)
	post	137.02 (50.06)	175.57 (48.42)	161.68 (74.36)	98.44 (40.75)	31.65 (15.02)	49.1 (15.5)	50.3 (10.1)	3.73 (1.23)	2.03 (0.93)
31–40 years	pre	147.41 (58.54)	229.66 (29.48)	189.97 (98.71)	141.61 (31.29)	36.37 (15.56)	38.6 (4.48)	58.7 (14.1)	4.60 (1.31)	2.89 (1.01)
	post	139.68 (53.35)	189.60 (40.64)	176.40 (94.97)	107.72 (36.06)	33.07 (15.60)	46.1 (10.4)	47.9 (10.8)	4.18 (1.36)	2.36 (1.00)
41–50 years	pre	150.18 (62.72)	232.28 (29.28)	207.05 (124.08)	139.40 (29.82)	36.54 (16.17)	39.9 (4.44)	57.4 (11.6)	4.67 (1.28)	2.87 (1.03)
	post	140.64 (50.94)	185.62 (40.46)	163.86 (89.86)	105.81 (34.39)	30.77 (13.70)	45.3 (11.1)	51.4 (12.5)	3.98 (1.24)	2.22 (0.90)
51–60 years	pre	151.55 (57.77)	231.95 (28.19)	204.57 (120.29)	140.07 (27.82)	36.38 (15.53)	38.1 (5.64)	61.2 (18.5)	4.66 (1.48)	2.92 (1.39)
	post	139.17 (52.35)	189.24 (38.78)	168.79 (96.36)	106.40 (32.93)	30.59 (13.13)	45.9 (11.3)	50.7 (12.0)	3.99 (1.15)	2.24 (0.94)
>60 years	pre	145.31 (59.76)	230.49 (26.58)	199.30 (113.65)	140.39 (29.50)	35.40 (15.03)	37.6 (6.3)	59.0 (13.8)	4.77 (1.26)	3.03 (1.45)
	post	138.32 (46.60)	182.81 (37.36)	169.85 (96.50)	103.92 (33.32)	32.01 (15.88)	47.5 (14.4)	46.9 (11.3)	4.11 (1.23)	2.33 (0.93)

**Table 2 medicines-06-00100-t002:** Shift in those with high lipid levels after yoga in diabetes patients.

Pre		Post				Sig *
DM (A1c > 6.5%)		Above Normal Range		Below Normal Range		
Variable	N	*n*	%	*n*	%	*p*
TC > 200 mg/dL	642	242	37.7	400	60.3	<0.001
LDL >130 mg/dL	392	103	26.3	289	73.7	<0.001
Tg > 200 mg/dL	433	160	37.0	273	63.0	<0.001
HDL < 45 mg/dL	835	470	56.3	365	43.7	<0.001

* McNemar’s test: *p* ≤ 0.001.

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
