# Peer review of "Efficacy of a Validated Yoga Protocol on Dyslipidemia in Diabetes Patients: NMB-2017 India Trial"

_medicines, 2019, doi:10.3390/medicines6040100_

Round 1

Reviewer 1 Report

This study using a stratified sample is a good effort to add to the body of knowledge related to the benefits of yoga. The study itself is scientifically sound, but the manuscript is not. Suggest combining 1.1 with 1.2. Suggest removing the reference to the AMA (20) recommendations for lifestyle changes and replacing it with one for the Indian population. There is an inconsistent use of acronyms and errors in the manuscript, i.e. p.2 line 58 "cardiocascular" instead of "cardiovascular." Many of the acronyms are not spelled out or not with the first use, i.e. "DYP." Suggest removing Table 1 and report the information within the text. Table 2 should be an appendix. The methods are not clearly explained and/or have inconsistent information. Suggest reporting the results in Table 4 with something other than "> or < normal. "There is a large drop-out rate, which is mentioned as a limitation only briefly. Likewise there are more limitations that should be added-lack of control of a participant's life (changes in diet, other exercise interventions, health status, etc.). 

Author Response

This study using a stratified sample is a good effort to add to the body of knowledge related to the benefits of yoga. The study itself is scientifically sound, but the manuscript is not.

Query 1: Suggest combining 1.1 with 1.2.

Answer: Sections have been combined 1.1 and 1.2 as suggested. See section 1.1; line number 33.

Query 2: Suggest removing the reference to the AMA (20) recommendations for lifestyle changes and replacing it with one for the Indian population.

Answer: AMA reference has been replaced with appropriate reference at the place. (Line number 70; reference 20)

Query 3: There is an inconsistent use of acronyms and errors in the manuscript, i.e. p.2 line 58 "cardiocascular" instead of "cardiovascular." Many of the acronyms are not spelled out or not with the first use, i.e. "DYP."

Answer: These inconsistencies have been corrected along with other changes. (Highlighted)

Query 4: Suggest removing Table 1 and report the information within the text.

Answer: Table 1 has been removed and as suggested added as text. Please see Line number 103-108.

Query 5: Table 2 should be an appendix.

Answer: Table 2 has been removed. Instead supplementary table 1 has been retained and our previous report has been cited.

Query 6: The methods are not clearly explained and/or have inconsistent information.

Answer: Our previous publication has been cited for detailed methodology (Reference no 26 & 27), Line number 90.

Query 7: Suggest reporting the results in Table 4 with something other than "> or < normal.

Answer: These symbols have been replaced Please see Table 2 now. line number 173.

Query 8: "There is a large drop-out rate, which is mentioned as a limitation only briefly. Likewise there are more limitations that should be added-lack of control of a participant's life (changes in diet, other exercise interventions, health status, etc.).

Answer: As suggested a few limitations have been mentioned. Please see line number238-240.

Reviewer 2 Report

This is a well written manuscript, employing appropriate methodology that would contribute to the literature.

Author Response

Thank you for your valuable inputs.

Reviewer 3 Report

Please provide list of abbreviation

Author Response

Comment: Please provide list of abbreviation

Answer: List of abbreviations has been provided.  Please see 262-276